# Exploring Pharmacy Students’ Perceptions of Feedback and Self-Reflection in Patient Counselling Simulations: Implications for Professional Development

**DOI:** 10.3390/pharmacy13030074

**Published:** 2025-05-27

**Authors:** Jessica Pace, Andrew Bartlett, Tiffany Iu, Jessica La, Jonathan Penm

**Affiliations:** 1Faculty of Medicine and Health, Sydney Pharmacy School, The University of Sydney, Sydney, NSW 2006, Australia; andrew.bartlett@sydney.edu.au (A.B.); tiiu2693@uni.sydney.edu.au (T.I.); jela9323@uni.sydney.edu.au (J.L.); jonathan.penm@sydney.edu.au (J.P.); 2Department of Pharmacy, Prince of Wales Hospital, Randwick, NSW 2031, Australia

**Keywords:** patient counselling, feedback, self-reflection, peer assessment, self-assessment

## Abstract

(1) Background: Structured use of feedback and self-reflection in simulated counselling sessions has a number of benefits, including identification of strategies for improvement, improvement in key skills and adaptability and a patient-centred approach which will help them to succeed as effective healthcare practitioners. The aim of this study was therefore to explore students’ perceptions of self-reflection and feedback in patient counselling simulations and the development of patient counselling skills; (2) Methods: Focus groups explored student perceptions of how the combination of self-reflection, self-assessment and teacher and peer feedback impacted their performance in simulated patient counselling assessments; (3) Results: Four focus groups with 21 pharmacy students were conducted. We identified three main themes and associated subthemes: consistency and continuity (sub-themes learning through repetitive assessment and inconsistent expectations), perceptions of feedback (sub-themes tutor feedback, peer feedback and self-reflection) and real-life practice (sub-themes authenticity of simulation cases, perceptions of empathy and professional development); (4) Conclusions: This study highlights the critical role of integrating consistent, high-quality feedback, peer assessment, and self-reflection in pharmacy education to enhance students’ learning experiences and prepare them for professional practice. As workplace-based assessment becomes more common and expected by accreditation bodies, these insights underscore the need for structured and continuous feedback processes to be integrated into all areas of pharmacy curricula.

## 1. Introduction

Building professional identity in pharmacy education is cultivated through the lenses of self-determination theory (SDT) and reflection. As Denys et al. state, “The essence of being a professional is the autonomous person who thrives in an environment of freedom, convenience and striving for self-determination as a means of personal and professional growth” [1]. Pharmacy students navigate a dynamic interplay between personal aspirations and the realities of practice as they transition to pharmacists, making it crucial to understand how to effectively support their development [2]. Mylrea et al. [3] highlighted that when students feel competent, connected, and autonomous, they develop a stronger professional identity. Through this, students can recognise their own unique skills and areas of improvement, making the transition to future practice less complex. Developing professional identity necessitates educational strategies, including self-reflection, peer feedback and tutor feedback, which cultivates autonomy, reflection and intrinsic motivation, aligning with SDT [4]. Self-reflection enhances autonomy by encouraging intrinsic motivation, allowing students to develop personalised learning strategies for professional growth. Competence is also reinforced through tutor feedback, as tailored feedback helps students refine their skills and provide guidance to future responsibilities in healthcare [5]. Peer feedback promotes relatedness, as it enables a collaborative learning environment with less academic pressure and meaningful feedback exchanges [6]. These tools cultivate a mindset of continuous improvement in students and also develop introspective learning as students recognise knowledge gaps to thrive as a future healthcare provider. Structured use of feedback and reflection encourages students to move beyond technical expertise and develop the adaptability and patient-centred approach required to thrive as effective healthcare practitioners [7].

Simulated counselling practice is widely implemented in pharmacy education as it supports pharmacy students’ formation of professional identity and develops critical skills; methods used include role-play with simulated patients or actors [8,9,10,11,12], peers [13,14,15,16,17] and faculty members [18,19,20,21], video recording of simulated communication [13,17,22,23], high-fidelity simulation [24,25], and mystery shoppers [26]. Patient counselling simulation enables students’ self-discovery by enhancing their practical skills, communication abilities and overall understanding of their future roles in healthcare [10,11,12]. Studies show that not only does this provide students with essential knowledge and skills but also fosters personal growth as they navigate pharmacy scenarios [13,15,16,17,27]. Furthermore, engaging in challenging interpersonal scenarios can help students develop critical soft skills that are essential for effective patient care [28,29]. This is critical for students to envision their future roles and their impact in the healthcare field [5,9]. Reflective tools and habitual self-assessment are key components of simulation-based tasks, as they stimulate deeper engagement in experiential learning [4,30]. Experiential learning theory emphasises the need to reflect on success and challenges in experiences in order for new insights to be applied to newer or similar tasks [31]. Self-assessment and feedback from various sources can not only give critical feedback on students’ performance and identify areas of improvement, but also help students develop a proactive approach to learning [32,33]. Providing students the autonomy to self-reflect in all aspects after simulated tasks, students can discover strategies to improve and deliver effective patient counselling, which motivates them to take greater ownership in their learning and have the desire to excel. The iterative process of counselling practice and the use of feedback loops allows students to apply feedback in subsequent scenarios, reinforcing their skills and ensuring continuous improvement. Overall, these experiences prepare students for their dynamic and multifaceted roles they will assume as pharmacists, and equip them with the skills and self-awareness necessary to thrive in real-world practice. 

Given the demonstrated benefits on intrinsic motivation and self-directed learning that self-assessment, peer feedback and tutor feedback can bring, the effectiveness of these tools towards learning in patient counselling simulation is also investigated in research. Studies have shown that roleplaying with peers and giving feedback drives observational learning, whereby students can learn from each other’s counselling performance and identify areas of improvements from within [13,14,15,16,17,33]. Common patterns through students’ perspectives shows that the reliability of feedback is largely valued, and that students tend to over criticise their own performance, highlighting the need for multiple perspectives [32]. While current research highlights the effectiveness of dynamic feedback mechanisms towards learning in simulation tasks, its use in combination with reflective practices and impact on student learning and development of patient counselling skills has not been explored in any depth. The aim was therefore to explore students’ perceptions of self-reflection and feedback in patient counselling simulations and the development of patient counselling skills.

## 2. Materials and Methods

This study included focus groups with pharmacy students about the use of self-reflection and peer assessment at simulated counselling sessions on their learning experiences. Ethics approval for this project was received from the University of Sydney’s Human Research Ethics Committee (protocol number 2018/603).

### 2.1. Study Setting

All third-year students in the University of Sydney Bachelor of Pharmacy degree are required to enrol in PHAR3825 Pharmaceutical Skills and Dispensing B. This unit of study consolidates previous units in the curriculum, building upon their therapeutic knowledge to counsel patients on the appropriate use of prescribed medications. In first- and second-year, students participate in simulated patient counselling interactions with their peers on over-the-counter (OTC) medicines. Over the course of the semester, students complete five dispensing and counselling sessions, focussing on 28 different simulated patient cases (four to six cases per session). In each of the sessions, students practice each of that week’s cases with a peer. This means that over the course of the semester, each student both completes and observes another student completing each of the 28 cases. Additionally, each student completes seven simulated patient counselling scenarios with a tutor; these are drawn from the 28 cases outlined above. Each session, students are also required to make a video of themselves counselling a peer on one of the cases and upload this to the unit’s Canvas Learning Management System (LMS) site. The tutor plays the role of the patient and marks the student on their performance in the simulated encounter. Both students and tutors provide feedback using a standard grading rubric (see Appendix A). This includes criteria related to both counselling content (e.g., the pharmacist introduces themselves and the purpose of the counselling session; discusses the name, purpose, dose, benefits and adverse effects of the medicine; and addresses patient concerns and provides a summary of key counselling points) and communication skills (e.g., demonstrates empathy and understanding, uses appropriate non-verbal communication and lay language and delivers information in manageable amounts). Students must be marked as satisfactory for all criteria on the grading rubric to successfully complete the case.

Case studies were drawn from the top 50 Pharmaceutical Benefits Scheme (PBS; Australia’s national health insurance scheme for medicines) dispensed drugs by volume [34] and students were provided with a standard structure to use when counselling patients, whereby counselling is based around “three prime questions”—(1) “what did the doctor tell you this medicine is for?”, (2) “how did the doctor tell you to take the medicine?” and (3) “what did the doctor tell you to expect?” [35].

In the initial iterations of the unit, students could choose which peer they completed and observed the case studies with (i.e., peer evaluations were identified and not assigned) and there was no screening of comments before release. However, this aspect was redesigned in 2022 on the basis of a previous course evaluation. Currently, the video that students submit to the LMS is randomly assigned to another student to mark using a provided grading rubric and feedback is provided anonymously to the student with feedback and comments screened before release. Students also mark their own performance using the standard grading rubric and self-reflect on their cases to identify their own unique learning needs. Self-reflection occurs both by completing a standard set of reflection questions after each counselling session and by completing a reflective statement based around a standard set of reflection questions at the end of the unit (see Figure 1). The learning outcomes primarily included effective counselling and education of patients about medicines and disease states.

Students also completed other activities in the unit in addition to the simulated patient counselling sessions. To ensure that they were adequately prepared for the simulated patient counselling sessions, students were required to complete a table summarising key information about each medication they were counselling on (such as indications, dose, precautions, side effects and storage) and submit this to the LMS before each session. In order to mimic the dispensing process in a community pharmacy, students also used FRED proprietary dispensing software to produce dispensing labels for each of the medications in the simulated patient counselling cases. Finally, they also completed a pharmaceutical science group project. Here, they analysed a drug using lab techniques such as UV spectrometry and high performance liquid chromatography (HPLC) and presented their findings in a report similar to those submitted by pharmaceutical companies to medicines regulatory bodies such as Australia’s Therapeutic Goods Administration (TGA). In order to align dispensing and counselling activities with the content covered in students’ other clinical units, sessions were timetabled so that students alternated between pharmaceutical science and dispensing activities each week.

Each year, there are around 210 third year Bachelor of Pharmacy students enrolled in this unit of study, with the majority being in their early 20s. All enrolled students were invited to participate in the focus groups via announcements in class and the unit Canvas learning management system site. These invitations included a written participant information statement informing students about the study (e.g., what was required, how long it would take, any risks and benefits and where to obtain further information). Interested students were invited to contact the research team and a suitable time for focus groups was arranged. Students provided written consent to participate in the focus group. There were no dropouts, and it was envisaged that up to 25 students would participate. As per previous studies [36], this is a well-accepted sample size in qualitative studies and is sufficient to achieve thematic saturation and therefore adequately explore variation in experiences amongst participants.

### 2.2. Data Collection

Four focus groups were conducted with 21 students (10 male and 11 female) in October 2022 by LM and LP, female interviewers and experienced pharmacy practice academics with qualitative research training and experience running focus groups and who were not teaching into this unit of study. Each researcher conducted two of the four focus groups. However, students may have had prior relationships with these academics from other units in their degrees. Focus groups lasted for an average of 25 min (range 15 to 30 min) and were conducted face-to-face at the University of Sydney campus. Only the participants and researchers were present for focus groups. Focus groups explored student perceptions of how this course and way of learning has impacted on their individual learning outcomes and preparedness to be a pharmacist. All focus group interviews were audio recorded (with the participants’ permission) and transcribed verbatim using Otta.ai transcription software (Otta.ai (https://otter.ai/), Mountainview, CA, USA); field notes were also taken. See Appendix A for focus group interview guide. The interview guide was developed after discussion by the academic team as an evaluation of the course and did not use a specific framework.

### 2.3. Data Analysis

Data were analysed via thematic analysis using an inductive approach. Data were checked for accuracy and entered into QSR International Nvivo software version 12 for data management and analysis. Each participant was assigned a number for anonymity. A phenomenological approach [37] underpinned the data analysis. This seeks to describe the essence of a phenomenon by exploring it from the perspective of those who have experienced it, with the aim of describing the meaning of the experience, both in terms of what was experienced and how it was experienced.

Data were analysed using iterative inductive thematic analysis procedures as outlined by Braun and Clarke [38]. This method of analysis is used for identifying, analysing and reporting patterns from data sources and developing interpretations of those patterns. Initial data coding and analysis were conducted by JL along with discussion amongst the authors to refine codes during this interpretive process [39]. This allowed the authors to observe significant patterns in the data, enabling an understanding of commonalities in participants’ responses. The analysis process involved six steps. First, all members of the research team familiarised themselves with data, e.g., by listening to recordings and reading transcripts. Initial codes were then generated by JL and reviewed by AB, JLP and JP. Ongoing discussion amongst JP, AB, JLP and JL was used to refine the code, which JL then used to code the code of the transcripts. Themes were then developed, defined and named themes through discussion amongst JP, AB, JLP and JL. Data analysis continued until all transcripts were analysed and thematic saturation (the point at which no new themes are emerging, and all themes are complete and well-described) [40] was reached. TI and JLP produced the final report which was reviewed and approved by all authors.

Although participants had the opportunity to review findings, none requested this and so no feedback was given.

We followed the Consolidated Criteria for Reporting Qualitative Research (COREQ) checklist [41] when reporting this study.

## 3. Results

We identified three main themes and associated subthemes. These are consistency and continuity (sub-themes learning through repetitive assessment and inconsistent expectations), perceptions of feedback (sub-themes tutor feedback, peer feedback and self-reflection) and real-life practice (sub-themes authenticity of simulation cases, perceptions of empathy and professional development). These are outlined in Table 1 and described in more detail below.

### 3.1. Consistency and Continuity

#### 3.1.1. Learning Through Repetitive Assessment

Students expressed a desire for more frequent counselling sessions, noting that they were unable to apply feedback and reinforce what they had learned due to the alternating classes that only allowed counselling practice to occur every two weeks. They outlined the alternating classes broke their momentum of improvement and shifted their focus. While the students were able to improve on one aspect, the break in learning was significant enough to cause lapses in other aspects.


*I noticed that because we did every two weeks, I felt like I was able to implement one thing, but then maybe I kind of fell off on something I was good on the week before.*



*It’s just by the time I’ve moved on to drug profiling that happens in our second week, my focus isn’t on counselling anymore until I start again, doing the next ones. But then I’ve forgotten about them.*


As such, weekly classes would allow students to maintain a more consistent and strong learning momentum where they would be able to apply provided feedback routinely and work to close gaps in their learning.


*I think that if we had a counselling session every week instead of every two weeks, then you could probably, by the end of week 12, you’re like, Ah, I’ve perfected this really good.*



*By doing it every week, and week in week out, I think kind of does get repetitive and like, you know, reflection is really good to do. But if you’re doing it every week, I feel like I got into habit of like, yeah, I did good in this. And then I could improve in this.*


#### 3.1.2. Inconsistent Expectations

Students found inconsistencies in expectations between tutors, notably in the types of questions asked, the amount of prompting that occurs prior to the assessments, and access to notes. These perceived inconsistencies affected their preparedness and performance, subsequently determining how the students were marked.


*My first concern is the tutors having different expectations. They all like a different style, which is fine. But then when you’re getting marked, some have different expectations, compared to others.*



*Different tutors will kind of go off script and ask you about different things. And if they’re going to not pass you because you didn’t mention that, then it’s unfair.*



*I had the experience where they basically prepped me before I started, which was great. They were like, I would like you to make sure that you ask about this, this and that. And I was like, that was good. But also at the same time, I was like, Why did you do that when no one else did that.*



*I guess when it’s marked, you don’t want to be marked down because you someone likes it a different way.*


### 3.2. Perceptions of Feedback

#### 3.2.1. Tutor Feedback

Students identified that while peer feedback was more valuable than self-reflection, the gold standard was tutor feedback. Students noted that tutor feedback was both supportive and reliable, as they were more experienced.


*I think the tutor feedback was the gold standard of what we got, that was the stuff I learned the most from. And then everything else, I don’t know, if I picked up anything from the other stuff.*



*I think the tutor’s feedback was good, too, in a sense that they didn’t just criticise what you were doing. They supported your skills, too.*



*I feel like the feedback that we were actually more concerned about is when it comes from the tutors, or, like, say Jessica (the unit of study coordinator) or someone, because they kind of just give you what we should have, because they have more experience.*


Additionally, tutor feedback was preferred as students were able to immediately apply the feedback to their second round of counselling practice.


*I prefer it from the tutors. And I guess I knew that also relates to me wanting it in the moment because write it down when they tell me it. Then we get to practice two in the session, so I can go back and apply it straightaway to the next one.*


However, as noted above, inconsistencies in expectation between tutors sometimes limited the usefulness of tutor feedback.

#### 3.2.2. Peer Feedback

Students considered peer feedback more useful than self-reflection, and appreciated the anonymity and usefulness of both the feedback itself and observing how other peers counselled.


*I really liked how we set it up, especially given that we made an anonymous marking setup, where I would mark and reflect on my work, someone or like another peer would mark and reflect on my work too, which is good to gain like another perspective.*



*I guess you get a second opinion. It will help you grasp if you didn’t get something the first time. Maybe you missed it again in your video as well. You pick up other people’s techniques as well, like you see how they’re counselling, and you can implement it in like our own. If it’s not something we usually see.*


Many students appreciated the anonymous aspect of peer feedback, noting that it encouraged honest and constructive criticism, as they were provided with a ‘veil’ to evaluate their peers. However, students noted that the quality of the feedback depended on the marker itself, expressing it was just a luck of the draw.


*There were a lot of constructive criticism given by our peers. And I think that really helped me to go through the course as well, just because I might have not been able to pick up some of the mistakes that I’ve made.*



*I think being anonymous helped with being a bit more harsh with the criticising. But I feel like that’s necessary. Just because if we just sugar-coat everything, we’re not gonna get through much.*



*I just think anonymous was a good idea. Because you really, like looked at the good parts and the bad parts. It gave you that veil to say, okay, you can improve on that, which is pretty cool. So yeah, I guess that was a really good part of the peer review process.*



*So it’s kind of a luck of the draw I guess. In terms of the feedback that is given, I don’t really have any really complaints about if its good, proper feedback.*


Despite these potential advantages, other students expressed that they did not look at or take the feedback seriously and did not find this valuable to their learning.


*Those videos were really bad. Really bad. Really dumb. Super pointless. At least for me, I just winged it and I’m pretty sure everyone who I commented on got a one word like comment. Because, it doesn’t help and I’m pretty sure most people don’t look at it. At least I didn’t.*



*The tutor’s feedback has helped me. The peer feedback I did not look at, honestly.*


#### 3.2.3. Self-Reflection

Some students found self-reflection to be useful initially, but expressed that its value decreased as they continually reinforced the learning over time. Additionally, students noted that self-reflection impacted their perception of self. They did not enjoy viewing their own videos, and preferred their performance being evaluated against the marking rubric by others.


*I think I found that in my first week, I got my marked video and then my self-reflection. So I looked at what I could have done better. But then as I improved on that, I felt as though it kind of dropped off in terms of usefulness, because I was applying what I had been a bit critical of in the first place. So by the time we got to like schedule five, I felt like I didn’t have a lot to say, because I was applying that throughout.*



*I don’t really want to watch myself do it back, I just didn’t find it helpful. I prefer the feedback when I’m doing the session, whether I’d be from our peers or demos because they’re going through the rubric.*


Many students also noted the nature of self-reflection is enacted instinctively. Thus, the process of submitting and evaluating their own performance on Canvas is quite redundant and tedious. They noted that the counselling session is already marked by the tutors and was essentially considered an ‘extra job’ that students did not put any thought into.


*I think the act of self-reflecting is pretty good. But it also happens like sort of second nature. So once you finish counselling, you’re like, Oh, damn, I did this wrong and stuff. But I think having on Canvas, like something that we have to put in might be like, an extra step, which we might not need. Since we already just do it after we finished counselling anyways.*



*It’s just like a tedious thing you have to do on canvas. It was just like one extra step you have to do after class. I think once class is done, everything should be done.*


They expressed a desire for self-reflection to be more specific to the case, rather than an evaluation of performance.


*I think if it was a bit more specific, more so than just the whole holistic view on the reflection of how we performed. But like, have it more so specific on the case that we had, rather than just having us talk about what we did and how well we did it.*


### 3.3. Real-Life Practice

#### 3.3.1. Authenticity of Simulation Cases

Students expressed a desire for authentic approaches to real-life practice but had mixed views on whether this was established through the simulation cases. They outlined that access to resources would be valuable to replicate real-life practice and would shift the focus from memorisation to developing interpersonal communication skills that would improve the authenticity of the counselling interaction.


*I think, in more of a real world aspect, you do have information in front of you, in reality. So I mean, memorising is good, obviously. But if this counselling session is to be very relevant to real-life situations then I think having notes in front of you would be beneficial.*



*If we were in a real pharmacy setting, I think I could have been able to bring out a CMI, or other notes that would be both beneficial for me and patient while counselling. So I think to reflect more of a real world situation, that was something that we might have been able to bring along.*



*I can focus more on clarity and structure and other more important things in a patient interaction. And it’s less like, okay, come up on everything, do I remember everything. Oh what if I forget this or sort of if I forget that. Yeah, I think in a real world setting, memorisation is not the key focus, especially as a pharmacist.*



*I guess having that option would be nice. And I think the focus would have shifted less from memorising and more out of clarity, and empathy and, like more interpersonal.*


Additionally, students noted that the time constraints in real-life practice limited the amount of information they were able to provide to a patient.


*You don’t really get to go through all the counselling points with patients usually. So it is helpful as we learn more about the medications, but sometimes they don’t represent real-life. A lot of the time you don’t actually have, you know, in a real community pharmacy, that time to go through that with the patient.*



*I guess now I do know things that maybe I would mention to a customer if they came in briefly, but having that five minute conversation that we are taught to have at uni isn’t a thing. We don’t really get to do that in community. But that’s, I guess the way it goes. Can’t really change that, but it is good to still learn how to do it.*


Other students felt that the learning environment was an accurate representation of real-life practice. They also appreciated how they were able to consolidate their knowledge, as the drugs assessed are all commonly dispensed in real-life pharmacy practice.


*I think being marked in exam conditions was good as well. It helps us practice for the real world situation, especially when we don’t know which one we’re getting. So it’s not just like blurting out information, and saying what you can remember.*



*They just took a really realistic approach. When you look at all the cases, it’s something that you usually come across, especially in like a normal regular community pharmacy setting.*



*I feel like most of the drugs that you guys give us are pretty common in everyday, especially at work. So it helps with reinforcing all the counselling points when I’m working.*


#### 3.3.2. Perceptions of Empathy

Students had mixed views on the way empathy was enforced. Many appreciated how the tutors were supportive and encouraging of the way they showed empathy, prompting students to implement these skills in real-life practice.


*When I was doing my counselling session, one of the tutors actually complimented me on the way I was showing empathy. And I don’t think that’s something I realised myself. So when she did say that, I made sure I was more empathetic in person when I actually spoke to a patient and like, you know, sometimes acknowledging the fact that they’re going through something, it helped me like create that connection.*


Students outlined how practicing empathy benefited their real-life practice, allowing them to adapt their counselling style and build relationships with patients in their workplace.


*I think empathy also teaches you to adapt to the situation. Like it’s not always going to be like, Oh, I understand how you’re feeling because you haven’t lived that experience, then you won’t sound sincere to the patient.*



*I’ve noticed that I’m implementing empathy, they’re more trusting of you and it builds that relationship.*


However, some students developed a negative perception of empathy, noting that it felt inauthentic and forced.


*It’s too emphasised and they push it a bit too hard. It’s almost a bit unnatural. I think it’s also harder to be sincere because obviously, like it’s not the actual patient, it’s a tutor.*


#### 3.3.3. Professional Development

Students outlined many benefits of the counselling assessments to their professional development, notably in regard to their competence and confidence when counselling patients. Students appreciated how the simulation cases were structured, providing a scaffold to follow when counselling in the workplace. Students also noted that their knowledge of the counselling points for the commonly dispensed drugs had improved, enabling them to ask appropriate questions and present information to execute the scripting out process properly.


*This unit in particular has helped me ask the right questions and get proper answers, and like, has helped me structure and present all the information I need to.*



*I think it’s very beneficial for like, our professional development as well, like in the workplace, I found that I’m starting to apply more of what I learned from counselling at work. I’m actually scripting out properly.*


Many students acknowledged the practicality of the three prime questions, noting that they apply it in real-life practice as it simplifies the scripting in process.


*The three main questions that helped us and prompted us were actually really good. The what do you expect, instructions and all that was really good. That really helped us structure everything.*



*I actually do the three prime questions. The three prime questions were actually probably the most helpful thing.*


Furthermore, students acknowledged the beneficial impact this unit has had on their confidence in counselling patients.


*I think that it helps build confidence, especially because I’m really shy. So like, having been forced to do it helps build confidence in talking to people and talking to patients.*



*I also feel a lot more confident to talk to them. Even as simple as learning how to structure the very foundational stuff. Like, even if I don’t know that much about the drug, I can just ask did the doctor talk to you about it. So yes, I definitely think its impacted my work.*


## 4. Discussion

To our knowledge, this is one of a small number of studies to examine the combination of educator, self- and peer-assessment and self-reflection on simulated counselling activities in pharmacy students; this was achieved through a qualitative evaluation of third year pharmacy students’ perceptions of the impact of these strategies on their learning experience. Our results emphasise the need for consistency and continuity in learning, the impact of (perceived) variations in feedback provided and links between these learning activities and students’ real-life practice. Our findings have important implications for the use of these techniques in developing pharmacy students’ patient counselling skills.

A key finding from this study is that students value the consistency, quality and reliability of feedback received from both tutors and peers. Our participants identified tutor feedback as the gold standard due to its authoritative nature, specificity and alignment with professional expectations. As seen in other studies, students recognise that tutors have the authority of experience in the professional field and feedback training, and their tailored feedback is essential to thrive in future practice [42]. However, students noted inconsistencies in the expectations set by different tutors, which could impact their learning experience. Students’ perception of tutor feedback is influenced by various factors, such as the student-tutor relationship, good communication, and the tutors’ ability to aid the link between theoretical knowledge and practical application [43]. To address inconsistencies in expectation, reducing the rotation of tutors involved per student can be suggested for students to build a good rapport and communication with particular tutors over time. Studies have also suggested that setting clear, written expectations at the beginning of rotations can help align and ensure clarity in tutor feedback [42].

On the other hand, peer feedback is shown to have more value when received anonymously but presents challenges in consistency and quality. Research suggests that anonymity fosters more constructive and honest feedback, as students feel less pressure and are more willing to provide meaningful critiques [6]. The findings of this study also highlight the variability in peer feedback quality, which is shaped by both intrinsic and extrinsic motivations. In online peer feedback, student characteristics and learning environment, particularly motivational factors like satisfaction and self-efficacy, impact feedback quality [23]. Students who view peer feedback with higher perceived benefits to both their own and their peers’ learning are more likely to engage in peer feedback activities. One study highlights that students’ demographic backgrounds, academic experiences, and psychological traits can influence the effectiveness of peer feedback, ultimately shaping their engagement and learning outcomes [44]. Logistical factors, such as the timing of assignments can also impact the volume and quality of peer comments [45]. Storjohann et al. [46] addressed the variability in peer feedback by allowing students to challenge inaccuracies in peer assessments, ensuring that feedback is not only received but also regulated for fairness and consistency. This suggests an implementation of a feedback-on-feedback model, such as a rating system to rate the usefulness of the feedback received, or providing feedback justification to encourage accountability. Given these findings, ensuring structured feedback processes, such as setting clear expectations for tutors and refining peer feedback mechanisms, may help enhance the consistency and reliability of feedback. However, it is important to consider time and resource constraints when implementing these initiatives. Further research is crucial to evaluate their feasibility and resource allocation. Maintaining consistency in feedback amongst students while preserving individualised learning is essential to ensure that students can accurately identify areas for improvement, but also to support each students’ professional growth and learning journey. 

Lastly, students felt that video-based self-review and written self-assessment required additional effort, and, over time, they perceived it as less useful and unnecessary. While studies have investigated the use of technological tools to minimise effort and enhance interactivity, the root issue lies in students’ lack of habitual engagement with reflective practices, as well as the absence of immediate perceived benefits from self-reflection [47]. Additionally, the act of reviewing one’s own recorded performance can evoke feelings of vulnerability and anxiety, which may explain the unreceptiveness to this learning activity [48]. In the context of pharmacy education, scaffolding reflective skills in the earlier stages of the curriculum is crucial for reinforcing the value of self-reflection in counselling practice. This suggests that increasing the frequency of counselling opportunities, accompanied by self-reflection throughout the pharmacy curriculum, can help students recognise its impact on their performance in formative oral assessments. It is essential to consider the pharmacy curriculum as a whole, ensuring that structured reflection sessions are systematically integrated across multiple units rather than treated as isolated exercises [31]. Reflective abilities of students may also be overestimated, which could contribute to their reluctance toward self-reflection. An educational workshop that fosters reflection through group debriefs and providing students an example of a reflection piece is shown to enhance reflective skills and establish clear expectations [49]. Moreover, the importance of self-reflection in both education and future professional practice cannot be overstated, particularly for pharmacy students in shaping their professional identity. Educating students on the role of life-long self-reflection in professional growth and self-awareness is crucial in the evolving field of pharmacy [5]. To address the lack of structure in written self-reflection and support professional identity formation, incorporating a structured approach such as the IDEA format (Identify, Describe, Evaluate, and Action Plan) has been shown to guide pharmacy students in their reflective practice. This format not only provides clarity but also encourages the creation of actionable professional development plans, underscoring the significance of self-reflection in future pharmacy practice [50]. By addressing barriers to engagement and systematically embedding structured reflection practices, a learning culture where self-reflection becomes valued in learning and professional development can be fostered amongst pharmacy students.

### Limitations

As with all qualitative research, the generalisability of our results is a potential issue. This is particularly important if trying to apply these results to students in other health disciplines and with other degree structures (e.g., where communication is more embedded in the degree or there is a greater integration of university teaching and workplace learning activities). It is also possible that our sampling strategy resulted in recruitment of a particular “type” of student—someone who cares enough about this issue to give up a considerable period of time to participate and/or who has a particular vision about their education and what activities they do or do not find valuable. Additionally, it is possible that our results do not represent the “true” beliefs and values of our students. For example, students might not be telling the truth and, even if they were not being overtly deceptive, the social desirability bias [51] may have led them to state what they think the interviewer or other students in the focus group wanted to hear, rather than what they truly believe and value. Indeed, it is even possible for people to deceive themselves about what they “truly” believe and value. However, the fact that we discovered a rich range of opinion about the learning activities in this unit and that thematic saturation was reached (with all themes complete and well-described) suggests that these are not major issues. Finally, this study sought only the views of students. While this, in itself, does not diminish our findings, educational initiatives are best evaluated through a range of lenses [52]—including the student, teacher and other educators—and incorporating other views could lead to a richer evaluation here.

## 5. Conclusions

This study highlights the critical role of integrating consistent, high-quality feedback, peer assessment, and self-reflection in pharmacy education to enhance students’ learning experiences and prepare them for professional practice. The findings emphasise that tutor feedback is highly valued for its authoritative and specific nature, though inconsistencies in expectations can impact learning. While peer feedback is beneficial, mechanisms are needed to ensure its consistency and quality. Self-reflection, although initially perceived as burdensome, is essential for developing professional identity and should be systematically embedded throughout the curriculum.

As workplace-based assessment becomes more common and expected by accreditation bodies, these insights underscore the need for structured and continuous feedback processes to be integrated into all areas of pharmacy curricula. Developing habits of reflective practice will help prepare students for the continuous learning required to be independent professionals. By addressing these areas, pharmacy programmes can better prepare students to become independent and competent professionals, capable of delivering high-quality patient care.

## Figures and Tables

**Figure 1 pharmacy-13-00074-f001:**
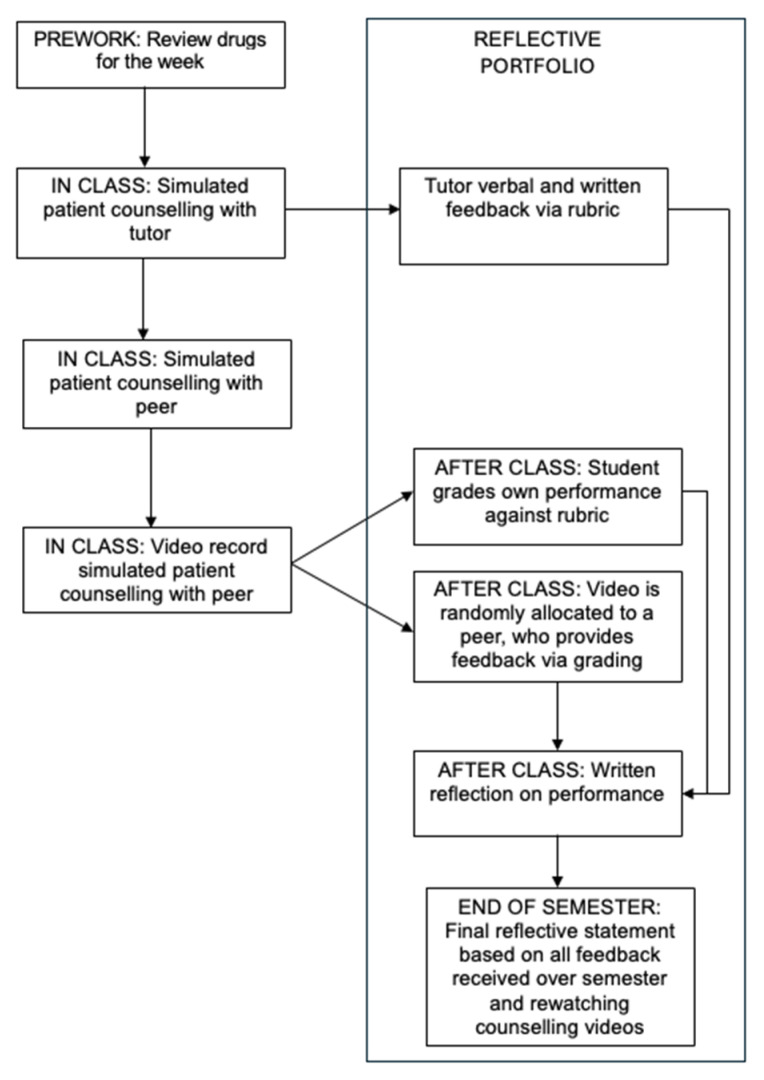
Structure of the learning activity.

**Table 1 pharmacy-13-00074-t001:** Overview of identified themes and subthemes.

Theme	Sub-Themes
Consistency and continuity	Learning through repetitive assessment
	Inconsistent expectation
Perceptions of feedback	Tutor feedback
	Peer feedback
	Self-reflection
Real-life practice	Authenticity of simulation cases
	Perceptions of empathy
	Professional development

## Data Availability

The raw data supporting the conclusions of this article will be made available by the authors on request.

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
