# Peer review of "Exploring Pharmacy Students’ Perceptions of Feedback and Self-Reflection in Patient Counselling Simulations: Implications for Professional Development"

_pharmacy, 2025, doi:10.3390/pharmacy13030074_

Round 1

Reviewer 1 Report

Comments and Suggestions for Authors

The research presents a worthwhile subject matter for pharmacy education of real relevance for professional practice and training. Authors thoroughly examine how different channels for feedback and self-assessment aid skill development by pharmacy students for counselling through the use of simulation settings. The theoretical framework is well-established, and the study design appears appropriate for addressing the research aims.

Author Response

Reviewer 1

The research presents a worthwhile subject matter for pharmacy education of real relevance for professional practice and training. Authors thoroughly examine how different channels for feedback and self-assessment aid skill development by pharmacy students for counselling through the use of simulation settings. The theoretical framework is well-established, and the study design appears appropriate for addressing the research aims.

We thank the reviewer for this feedback.

Reviewer 2 Report

Comments and Suggestions for Authors

This article presents a report of a qualitative study of student perceptions of feedback on their performance in patient counseling simulations, with data collected using student focus groups. The topic would be of interest to readers, and the study methods are sound. The writing is clear, and the organization of the paper is effective. However, there are important details missing in some areas, as outlined below.

Abstract:

  • Clearly and concisely written. One point of clarification: I am not familiar with the term “demonstrator” in the context of simulation. Is this synonymous with “facilitator”? What is the role of a demonstrator? It seems they may have served as the “patient” in the simulated encounter. This wasn’t clear throughout the article.

Introduction:

  • A concise, but effective overview of the existing literature on the topic; no suggestions for revision.

Materials and Methods:

  • (Lines 108-112) I read the sentence beginning with, “In 2018 . . .” several times and still could not completely understand its meaning. Perhaps a reword or breaking it into two sentences would make it clearer. So students observed 28 counseling scenarios to conduct peer reviews? Seems like a lot, and I did not glean this from the description of results.
  • Describe the role of the demonstrator for readers who are unfamiliar with this term.
  • (Lines 127-128) Is this sentence missing a word after “grading”? Should it be “rubric”? A brief description of the rubric would also help to better understand the nature of the feedback.
  • The approach to qualitative analysis seems sound, and the authors used multiple reviewers to analyze the data. The authors state in Line 184 that, “Data analysis continued until all transcripts were analyzed.” However, one thing that is unclear is whether the codes originally generated by JL then reviewed by others and refined were then used to go back through all the transcripts to code them, and if this phase was completed by more than one coder. If multiple coders were used, how were discrepancies in coding resolved?

Results:

  • In some places, components of the experience are mentioned in student quotes that are not described in Section 2.1, making it difficult to make meaning of the comments. For example, in Lines 206-207, “alternating classes” are mentioned as a hindrance to student retention/momentum. Consider explaining the alternating class structure in the description of the setting.
  • (Line 268) Who is “Jessica” who was mentioned in the student quote? Is she a demonstrator or teaching assistant? It would be helpful to put Jessica’s role in brackets so the reader can put the student feedback in context.
  • (Lines 345-347) The comment about students noting that self-reflection was instinctive, making repetitive assignments redundant, seems very similar to the next comment (Lines 359-363). Consider reworking these two to better distinguish them or consider combining the two.
  • (Lines 372-373) It isn’t clear what the student is referring to, as videos, labels, and drug tables were not mentioned as part of the course experience in Section 2.1.

Discussion:

  • (Lines 534-536) Regarding the sentence that starts, “In online peer feedback . . .”, in relation to the sentence before and after this one and the absence of a citation, it seems this statement is referring to the current study, not an existing study. In that case, how did the authors conclude from the findings that are presented that “ . . . motivational factors like satisfaction and self-efficacy impact feedback quality.” I didn’t see that finding mentioned in the results.

Limitations

  • No recommendation for revision.

Conclusions

  • No recommendation for revision.

Supplementary Materials:

  • A tracked changes version of the focus group guide is provided. Was this intentional? Does this represent the final version used by the researchers?

Author Response

Reviewer 2:

This article presents a report of a qualitative study of student perceptions of feedback on their performance in patient counselling simulations, with data collected using student focus groups. The topic would be of interest to readers, and the study methods are sound. The writing is clear, and the organization of the paper is effective. However, there are important details missing in some areas, as outlined below.

Thank you for this feedback. Please see below for our responses to the issues you have raised.

Abstract:

  • Clearly and concisely written. One point of clarification: I am not familiar with the term “demonstrator” in the context of simulation. Is this synonymous with “facilitator”? What is the role of a demonstrator? It seems they may have served as the “patient” in the simulated encounter. This wasn’t clear throughout the article.

We apologise for the confusion! A demonstrator is a tutor, who plays the role of the patient and marks the student on their performance in the simulated encounter. We have replaced the term demonstrator with tutor in the revised manuscript.

Introduction:

  • A concise, but effective overview of the existing literature on the topic; no suggestions for revision.

Thank you for this feedback.

Materials and Methods:

  • (Lines 108-112) I read the sentence beginning with, “In 2018 . . .” several times and still could not completely understand its meaning. Perhaps a reword or breaking it into two sentences would make it clearer. So students observed 28 counseling scenarios to conduct peer reviews? Seems like a lot, and I did not glean this from the description of results.

Sorry for the confusion! Over the five sessions held in the semester, the students looked at 28 cases (4 to 6 cases per session). In class time, students practiced each of these cases with a peer. This means that each student both completed a simulated role play of each case with a peer and observed a peer complete each case when they were playing the patient. They also recorded a video of themselves completing one of these cases with a peer and uploaded this to the online LMS. They were then allocated another student’s video to review. The cases that students completed with a tutor were also drawn from these 28 cases. Therefore, students completed and observed 28 cases with a peer in their class time and completed 5 peer reviews of another students counselling video over the course of the semester. This has been clarified in the revised manuscript.

  • Describe the role of the demonstrator for readers who are unfamiliar with this term.

As noted above, a demonstrator is a tutor, who plays the role of the patient and marks the student on their performance in the simulated encounter. We have replaced the term demonstrator with tutor throughout the revised manuscript and outlined the role that they play in these sessions in section 2.1 Study Setting (page 3 lines 118-120).

  • (Lines 127-128) Is this sentence missing a word after “grading”? Should it be “rubric”? A brief description of the rubric would also help to better understand the nature of the feedback.

Yes, the word rubric was missing here. We have added this and a brief description of the grading rubric to the revised manuscript (see page 3 lines 121-131). A copy of the grading rubric is also provided as part of the supplementary materials.

  • The approach to qualitative analysis seems sound, and the authors used multiple reviewers to analyze the data. The authors state in Line 184 that, “Data analysis continued until all transcripts were analyzed.” However, one thing that is unclear is whether the codes originally generated by JL then reviewed by others and refined were then used to go back through all the transcripts to code them, and if this phase was completed by more than one coder. If multiple coders were used, how were discrepancies in coding resolved?

Yes, the codes generated by JL were reviewed by all authors and refined and then used to go back through and code all of the transcripts. This phase was completed by a single coder JL. This has been clarified in the revised manuscript.

Results:

  • In some places, components of the experience are mentioned in student quotes that are not described in Section 2.1, making it difficult to make meaning of the comments. For example, in Lines 206-207, “alternating classes” are mentioned as a hindrance to student retention/momentum. Consider explaining the alternating class structure in the description of the setting.

This has been added (page 5 lines 168-182).

  • (Line 268) Who is “Jessica” who was mentioned in the student quote? Is she a demonstrator or teaching assistant? It would be helpful to put Jessica’s role in brackets so the reader can put the student feedback in context.

Jessica is the unit of study coordinator. Her role has been added to this quote.

  • (Lines 345-347) The comment about students noting that self-reflection was instinctive, making repetitive assignments redundant, seems very similar to the next comment (Lines 359-363). Consider reworking these two to better distinguish them or consider combining the two.

These two comments have been combined (see page 10 lines 407-418).

  • (Lines 372-373) It isn’t clear what the student is referring to, as videos, labels, and drug tables were not mentioned as part of the course experience in Section 2.1.

This has been added to Section 2.1 (page 3 lines 119-121 for videos and page 5 lines 168- for labels and drug tables).

Discussion:

  • (Lines 534-536) Regarding the sentence that starts, “In online peer feedback . . .”, in relation to the sentence before and after this one and the absence of a citation, it seems this statement is referring to the current study, not an existing study. In that case, how did the authors conclude from the findings that are presented that “...motivational factors like satisfaction and self-efficacy impact feedback quality.” I didn’t see that finding mentioned in the results.

This statement refers to an existing study—Kerman T, Banihashem N, Kazem S, et al. Online peer feedback in higher education: A synthesis of the literature. Education and Information Technologies. 2023;29:1-51. doi: 10.1007/s10639-023-12273-8. We have added the citation to this sentence.

Limitations

  • No recommendation for revision.

Conclusions

  • No recommendation for revision.

Supplementary Materials:

  • A tracked changes version of the focus group guide is provided. Was this intentional? Does this represent the final version used by the researchers?

This was an oversight on our part and was not intentional. This was the final version used by the researchers and a clean copy of the focus group guide has been provided with the revised paper.

Reviewer 3 Report

Comments and Suggestions for Authors

Thank-you for the opportunity to review this paper. Feedback and self-reflection are indeed important for pharmacy student development, particularly in the area of patient counselling. This paper is quite well written and will be of high interest to pharmacy educators.

I have several comments:

  1. Introduction - the introduction to the paper adequately covered the background information for the study but could be referenced more completely, particularly lines 63, 66, 72 and 85. For lines 63 and 85, you mentioned ‘studies’ and quoted only one study.

There was also some repetition in the text from the abstract – lines 11 and 13 (abstract) and lines 55 and 56.

In line 35 you mentioned SDT, reflection and autonomy as the processes to build professional identity – as autonomy is part of SDT, I feel that this does not need to be included here.

  1. Materials and Methods
  • While your title includes student perceptions of demonstrator feedback, I noticed that this is not mentioned as part of the focus group interview guide. Perhaps you could explain how this integrates with the study outcomes?

Methods seem to be appropriate and described in good detail – however, several areas were unclear.

  • Were these face-to-face or online interview? I am assuming that on campus means that they were face-to-face?
  • Was written information provided prior to the focus groups and was written consent obtained from the students?
  • I note that two researchers conducted the focus groups – were both involved in the questioning or was one the interviewer and one the observer? Were the roles consistent for each focus group?
  1. Results

Results were presented well – however, you mentioned that participants were identified by a number – it would be good to have this number beside each quote and perhaps whether they were male or female (if this was recorded and if you think it has significance)       

Author Response

Reviewer 3:

Thank-you for the opportunity to review this paper. Feedback and self-reflection are indeed important for pharmacy student development, particularly in the area of patient counselling. This paper is quite well written and will be of high interest to pharmacy educators.

I have several comments:

1. Introduction - the introduction to the paper adequately covered the background information for the study but could be referenced more completely, particularly lines 63, 66, 72 and 85. For lines 63 and 85, you mentioned ‘studies’ and quoted only one study.

Additional references have been added to the introduction.

There was also some repetition in the text from the abstract – lines 11 and 13 (abstract) and lines 55 and 56.

The text in the abstract has been revised (page 1 lines 10-13).

In line 35 you mentioned SDT, reflection and autonomy as the processes to build professional identity – as autonomy is part of SDT, I feel that this does not need to be included here.

Autonomy has been removed here.

2. Materials and Methods

  • While your title includes student perceptions of demonstrator feedback, I noticed that this is not mentioned as part of the focus group interview guide. Perhaps you could explain how this integrates with the study outcomes?

The aim of the study was to explore students’ perceptions of self-reflection and feedback patient counselling simulations and the development of patient counselling skills. When we wrote the interview guide, we wanted to focus on peer feedback; however, students also raised the importance and issues related to demonstrator feedback in the focus group discussions and this became important in our data analysis. We have adjusted the title and study aims in both the abstract and introduction to reflect this and better match the focus group interview guide.

Methods seem to be appropriate and described in good detail – however, several areas were unclear.

  • Were these face-to-face or online interview? I am assuming that on campus means that they were face-to-face?

Interviews were conducted face-to-face—this has been added to the revised manuscript (page 5 line 201).

  • Was written information provided prior to the focus groups and was written consent obtained from the students?

Yes, the invitation to participate included a written participant information statement informing students about the study (e.g. what was required, how long it would take, any risks and benefits and where to obtain further information). Students who agreed to participate provided written consent. This has been added to the revised manuscript (page 5 lines 189-191 and 193).

  • I note that two researchers conducted the focus groups – were both involved in the questioning or was one the interviewer and one the observer? Were the roles consistent for each focus group?

Two researchers conducted the focus groups, but each focus group only had one researcher (i.e. the researchers conducted two of the four focus groups each). This has been clarified in the revised manuscript (page 5 lines 203-204). 

3. Results

Results were presented well – however, you mentioned that participants were identified by a number – it would be good to have this number beside each quote and perhaps whether they were male or female (if this was recorded and if you think it has significance)  

It is true that all participants were assigned a number as part of the consent process and organisation of focus groups. However, we have reviewed the transcripts and recordings and are not able to match individual quotes to participant numbers or demographics. Therefore, while we agree this would be useful information to include, we were not able to include participant numbers and genders here.